# Disulfiram and copper combination therapy targets NPL4, cancer stem cells and extends survival in a medulloblastoma model

Riccardo Serra[1]*, Tianna Zhao[1], Sakibul Huq[1], Noah Leviton Gorelick[1], Joshua Casaos[1], Arba Cecia[1], Antonella Mangraviti[2], Charles Eberhart[3], Renyuan Bai[1], Alessandro Olivi[2], Henry Brem[1,4,5,6], Eric M. Jackson[1], Betty Tyler[1]

1 Department of Neurosurgery, Hunterian Neurosurgical Research Laboratory, Johns Hopkins University School of Medicine, Baltimore, Maryland, United States of America, 2 Department of Neurosurgery, School of Medicine - Catholic University of the Sacred Heart, Rome, Italy, 3 Department of Pathology, Johns Hopkins University School of Medicine, Baltimore, Maryland, United States of America, 4 Department of Biomedical Engineering, Johns Hopkins University School of Medicine, Baltimore, Maryland, United States of America, 5 Department of Oncology, Johns Hopkins University School of Medicine, Baltimore, Maryland, United States of America, 6 Department of Opthalmology, Johns Hopkins University School of Medicine, Baltimore, Maryland, United States of America

* rserra@som.umaryland.edu

**Data Availability Statement:** All relevant data are within the manuscript and its Supporting information files.

## Abstract

### Background

Medulloblastoma (MB) is the most common brain malignancy in children, and is still responsible for significant mortality and morbidity. The aim of this study was to assess the safety and efficacy of Disulfiram (DSF), an FDA-approved inhibitor of Aldehyde-Dehydrogenase (ALDH), and Copper ($Cu^{++}$) in human SSH-driven and Group 3 MB. The molecular mechanisms, effect on cancer-stem-cells (CSC) and DNA damage were investigated in xenograft models.

### Methods

The cytotoxic and anti-CSC effects of DSF/$Cu^{++}$ were evaluated with clonogenic assays, flow-cytometry, immunofluorescence, western-blotting. ONS76, UW228 (SHH-driven with Tp53m), D425med, D283 and D341 (Group 3) cell-lines were used. *In vivo* survival and nuclear protein localization protein-4 (NPL4), Ki67, Cleaved-Caspase-3, GFAP and NeuN expression were assessed in two Group 3 MB xenografts with immunohistochemistry and western-blotting.

### Results

Significant *in vitro* cytotoxicity was demonstrated at nanomolar concentrations. DSF/$Cu^{++}$ induced cell-death through NPL4 accumulation in cell-nucleus and buildup of poly-ubiquity-lated proteins. Flow-cytometry demonstrated a significant decrease in $ALDH^+$, $Nestin^+$ and $CD133^+$ following treatment, anti-CSC effect was confirmed *in vitro* and *in vivo*. DSF/$Cu^{++}$ prolonged survival, and increased nuclear NPL4 expression *in vivo*.

**Funding:** This work was supported by Ms. Kimberly Spiro, the Donald W. Spiro Foundation, and Donald R. Spiro.

**Competing interests:** The authors have read the journal's policy and have the following competing interests: Henry Brem is a paid consultant to Insightec and chairman of the company's Medical Advisory Board. Insightec is developing focused ultrasound treatments for brain tumors. This arrangement has been reviewed and approved by Johns Hopkins University in accordance with its conflict-of-interest policies. Dr. Brem also receives research funding from NIH, Johns Hopkins University, and Acuity Bio Corp* and philanthropy unrelated to this study. Dr. Brem is also a consultant for AsclepiX Therapeutics, StemGen, InSightec*, Accelerating Combination Therapies*, Catalio Nexus Fund II, LLC*, LikeMinds, Inc*, Galen Robotics, Inc.* and Nurami Medical* (includes equity or options). Betty Tyler has research funding from NIH and is a co-owner for Accelerating Combination Therapies (*includes equity or options). Ashvattha Therapeutics Inc. has also licensed one of Dr. Tyler's patents [Patent number: 8895597]. This does not alter our adherence to PLOS ONE policies on sharing data and materials.

## Conclusions

Our data suggest that this combination may serve as a novel treatment, as monotherapy or in combination with existing therapies, for aggressive subtypes of pediatric MB.

## Background

Medulloblastoma (MB), the second most common malignant brain tumor in the pediatric population, has a bimodal peak incidence between 3–4 and 8–10 years of age [1]. Its prognosis is generally poor and correlated with several factors, including age at diagnosis, extent of surgical resection, molecular subtype, and presence of metastases [1–3]. Patients typically receive multi-agent chemo- and radiation therapy with the aim of improving survival and diminishing radiation exposure in young patients [4–12]. Among the four main molecular subtypes, Shh-driven (with mutated Tp53) and Group 3 have the worst 5-year survival rates: 41% and 50%, respectively. Consequently, extensive research has focused on improving the outcomes of patients with these tumors [2]. However, current therapies have not achieved a substantial increase in median survival, and further research is needed to introduce new strategies aimed at limiting the toxicity of radiation and chemotherapy in this pediatric population.

Disulfiram (DSF), a drug used for decades for the treatment of chronic alcoholism, has recently shown significant cytotoxicity against a wide range of malignant cancers, especially when combined with copper gluconate (Cu$^{++}$) [13–21]. DSF, primarily an inhibitor of the enzyme aldehyde dehydrogenase (ALDH), achieves its anti-tumor effect by targeting several other intracellular pathways, including Nuclear Factor kappa-light-chain-enhancer of activated B-cells (NF-κB), and nuclear protein localization protein-4 (NPL-4), a mediator of the p97 ATPase cascade with higher expression in medulloblastoma cells and in other cancers of CNS origin [19, 20]. Combination therapy with Cu$^{++}$, a safe and inexpensive ion that easily crosses the blood-brain barrier (BBB), further enhances the cytotoxicity of DSF, lowering the IC$_{50}$ to nanomolar levels without increasing the risk of additional toxicities [13, 14, 17, 20, 21]. DSF is metabolized to ditiocarb (diethyldithiocarbamate, DTC), which then chelates Cu$^{++}$ *in vivo* to form the active DTC-Cu$^{++}$ complex (bis(diethyldithiocarbamate)–Cu$^{++}$ (CuET)). CuET, the anti-cancer derivative of DSF, has been demonstrated to accumulate in the brain and tumor tissues at therapeutic concentrations after a single administration [20].

Despite recent evidence supporting p97 inhibition as the main target of DSF-Cu$^{++}$ cytotoxicity in cancer [20], a number of other studies suggested different mechanisms to explain their anti-cancer activity. Among them, the inhibition of ALDH (one of the main targets of DSF) and of DNA repair mechanisms have been thought to play a central role against cancer stem cells (CSC), in CNS and non-CNS malignancies [13–16, 19]. However, in recent years questions have arisen on the contribution of ALDH inhibition to DSF-mediated CSC toxicity. New evidence seems to support a reduction in ALDH that is only secondary to membrane damage and cell death, rather than a direct consequence of ALDH inhibition from DSF-Cu$^{++}$ [22]. In order to better define the effects on CSCs and provide a solid platform for future repositioning of DSF-Cu$^{++}$, cellular expression of ALDH and of other markers of cancer cell stemness (CD133 and Nestin) were included in our analysis using both *in vivo* and *in vitro* techniques.

Further, since previous evidence pointed toward DNA damage and repair mechanisms as other potential mediators of DSF-Cu$^{++}$ anti-cancer activity, our study also tried to encompass a number of targets involved in these pathways, such as H2AX, CHK-1 and phospho-CHK-1 [14].

Finally, we evaluated the *in vitro* and *in vivo* efficacy of DSF-Cu$^{++}$ in multiple preclinical models of Shh-driven and Group 3 MBs, and demonstrated their efficacy as NPL4 inhibitors [20]. We believe that by characterizing at the molecular level the main targets of DSF-Cu$^{++}$ in medulloblastoma, this study will not only introduce a novel repurposed chemotherapeutic agent for medulloblastoma, but also provide a solid base for future exploration of its clinical uses, alone or in combination with other chemoradiation regimens. The deciphering of these mechanisms will facilitate the repositioning of DSF-Cu$^{++}$ in pediatric MB, offering an FDA-approved tool for the treatment of this aggressive brain tumor.

## Materials and methods

### Cell cultures

MB cell lines were cultured at 37˚C and 5% $CO_2$ using MEM (D425med, D341, D283), RPMI (ONS-76), DMEM-F12 (UW-228) media (Gibco, Gaithersburg, MD), supplemented with 10% Fetal Bovine Serum (Sigma-Aldrich) and 1% Penicillin/Streptomycin. Neurospheres derived from D425med and D341 cell lines were cultured in B27-supplemented Neurobasal medium, EGF and FGF (Gibco, Gaithersburg, MD). Shh-driven (UW-228) and Group 3 MB lines (D425med, D283) were obtained from the Johns Hopkins University, D341 (Group 3) was purchased from ATCC (American Type Culture Collection) and ONS76 was purchased from JCRB Cell-Bank (Japanese Cancer Research Resources Bank). Cells were authenticated and tested for Mycoplasma infection with Hoechst stain.

### Cell viability assay

DSF and Cu$^{++}$ were purchased from Sigma-Aldrich, and dissolved in ethanol (Pharmco, Brookfield, CT, USA). Cell viability was assessed using Cell Counting Kit 8 (Dojindo Molecular Technologies, Japan). $2x10^3$ cells were seeded on a 96-well plate and treated after 24h with DSF, Cu$^{++}$ and an equimolar combination of DSF-Cu$^{++}$. CCK-8 reagent was added after 48h and cell viability assessed. The data were normalized against untreated controls.

### Proliferation assay

The proliferative capacity of MB cells was assessed by seeding $2.5x10^4$ cells/well in 6-well plates, treating cells after 24h with 150nM DSF or equimolar DSF-Cu$^{++}$, and counting them after 96-144h with a hemocytometer.

### Clonogenic assay

The clonogenic potential of UW-228 and ONS-76 cells was investigated by seeding 500 cells/well; cells were treated with DSF-Cu$^{++}$ 150nM and incubated for 7 days. Wells were fixed and stained with crystal violet (Sigma-Aldrich, St. Louis, MO, USA).

### Neurosphere formation assay

D425med and D341 lines were cultured in Neurobasal medium at a density of $2.5x10^5$/ml and treated with 150nM DSF-Cu$^{++}$. Neurospheres containing more than 50 cells were assessed after 72h with inverted microscopy, and images were acquired with a Zeiss Axiovert Microscope (Carl Zeiss, Oberkochen, Germany).

## Western Blot (WB) analysis

UW-228, ONS-76, D425med and D341 cells were treated with 150nM DSF/Cu$^{++}$/DSF-Cu$^{++}$ (for either 12h **Fig 2C**; or 24h **Fig 3 Panel B**), or with 500nM and 1μM DSF-Cu$^{++}$ for 24h (**Fig 3 Panel A**), and pellets were harvested and frozen. D425med and D341 cells were also treated with a p97-inhibitor, ML240 (MedChemExpress, Monmouth Junction, NJ, USA), at a concentration of 100nM for 24h. Brain tissue samples from *in vivo* studies (tumor and normal brain) were snap-frozen in liquid nitrogen, extracted with surgical instruments, and lysed with an electric homogenizer (anti-NPL4, anti-ALDH1, anti-AIF, anti-phospho-Histone H2A.X and anti-GADPH antibodies were used for *ex vivo* studies). Protein concentration was then determined with BCA Assay (GE Healthcare, Marlborough, MA, USA) and membranes blotted with anti-ubiquitin (1:1 000; Cell Signaling, Danvers, MA, USA), anti-NRF1 (1:1 000; Cell Signaling, Danvers, MA, USA), anti-NPL4 (1:1 000; Novus Bio), anti-PARP (1:1 000; Cell Signaling), anti-AIF (1:1 000; Cell Signaling), anti-phospho-Histone H2A.X (Ser139) (1:500, Merk-Millipore, Burlington, MA, USA), anti-Chk-1 (1:1 000; Cell Signaling), anti-Phospho-Chk-1 (1:1 000; Cell Signaling), anti-ALDH1 (1:1000; Cell Signaling), anti-β-Tubulin antibody (1:1 000, Cell Signaling), and anti-GAPDH (1:1000, Merk-Millipore, Burlington, MA, USA). Membranes were incubated with horseradish peroxidase-conjugated secondary antibody (1:5 000; Jackson Laboratory, Bar Harbor, ME, USA), and band density was quantified with ImageJ (National Institute of Health) and normalized to the β-tubulin control signal.

## Immunofluorescence

Cells were incubated in primary antibody (anti-NPL4, 1:1 000; Novus Bio; anti-AIF, 1:1 000; Cell Signaling; anti-phospho-Histone H2A.X (Ser139) 1:500, Merk-Millipore) after treatment and fixation, and stained with the secondary antibody (Jackson Laboratory, Bar Harbor, ME, USA), and DAPI (Invitrogen, Carlsbad, CA, USA). Slides were prepared with mounting medium (Dako, Santa Clara, CA, USA) and imaged with a Zeiss microscope. Signal intensity was quantified with ImageJ (National Institute of Health).

## Flow cytometry

Early (AnnexinV positive) and Late (AnnexinV-Propidium Iodide (PI) positive) apoptotic cells were assessed with flow cytometry after 12/24/48h from treatment. UW-228, ONS-76, D425med, D283 and D341 cells were seeded at a density of $5.0x10^4$ cells/well, and treated with DSF-Cu$^{++}$ 150nM and 2uM. Cells were trypsinized and harvested, washed with PBS, incubated with AnnexinV Ab and PI (Invitrogen, Carlsbad, CA, USA), and analyzed. Cell proliferation and cycle were measured by Ki67 and PI expression. Cells were seeded at $2.5x10^4$/well, treated after 24h with 150/300nM DSF-Cu$^{++}$, and harvested after 24/48h of treatment. Pellets were washed with PBS, fixed and permeabilized with TritonX-100, and stained with anti-Ki67 antibody (Invitrogen, Carlsbad, CA, USA) and PI/RNase solution (FxCycle™, Thermo-Fisher Scientific, Waltham, MA, USA). For cancer stem-cell (CSC) identification, anti-CD133 antibody (BioLegend, San Diego, CA, USA), anti-Nestin (BioLegend, San Diego, CA, USA), and ALDE-FLUOR kit (Stem Cell Tech., Durham, NC, USA) were used. Cells were prepared and analyzed with BD FACS Celesta (BD Biosciences; Becton, Dickinson and Company, San Jose, CA, USA), and data analyzed with FlowJo software (FlowJo, LLC).

## Intracranial orthotopic mouse model and safety studies

Animal experiments were carried out to establish toxicity and maximum tolerated dose of oral DSF/Cu$^{++}$, and to assess the survival of athymic Nu/Nu mice (Charles River Laboratories, MA,

USA) bearing D425med and D341 tumors. All animals were housed at the Johns Hopkins University animal facility, and were given free access to food and water. Animal experiments were approved by the Johns Hopkins University Animal Care and Use Committee (ACUC). For *in vivo* experiments animals were anesthetized with Xylaket (a mix of 100% ethanol, Ketamine, Xylazine and Normal Saline), and a 1mm burr-hole was drilled over the left parietal area (2mm lateral to the sagittal suture and 3mm posterior to the coronal suture) and $1.25 \times 10^5$/animal D425med and D341 cells were implanted intracranially in 16 and 12 mice, respectively. Animals were randomized and treated by oral gavage with DSF (150 mg/kg/day) and Cu$^{++}$ (2 mg/kg/day) diluted in water (controls n = 8; DSF-Cu$^{++}$ n = 8). DSF and Cu$^{++}$ were delivered separately, for 3 cycles with each consisting of 5 days of treatment followed by two nontreatment days. Animals were euthanized when they reached a humane survival endpoint, reflecting tumor-induced neurologic and/or behavioral signs (lethargy, ataxia, paralysis, hunched posture per protocol). Animals were checked twice daily and euthanasia was carried out when criteria were met with 100% $CO_2$ atmosphere or by means of a lethal intraperitoneal injection of Xylaket, following ACUC's guidelines (animals euthanized within one hour, no animals died before meeting criteria). Survival was assessed and autopsies performed to remove and process the brains.

Two additional experiments assessed DSF-Cu$^{++}$ safety *in vivo* in mice without intracranial tumors (n = 8 and n = 12). In the first study, animals received oral DSF (150 mg/kg/day) and Cu$^{++}$ (2 mg/kg/day) or oral gavage of water daily, while in the second experiment three groups were administered DSF-Cu$^{++}$ at different dosages (50-150-250 mg/kg/day), and one received oral water gavage. The DSF and Cu$^{++}$ dosages are similar to those tolerated in clinical trials, and significantly lower than the maximum tolerated dose in humans.

## Histopathological analysis

Brains of symptomatic animals were perfused with 4% paraformaldehyde (Sigma Aldrich) and switched to 10% formalin after 24h for long-term preservation. All samples were paraffin-embedded by the Johns Hopkins University Histopathology Core and stained with hematoxylin-eosin (Sigma Aldrich), and with immunohistochemistry for anti-NPL4 (1:100; Novus Bio), Ki67 (1:100; Cell Signaling), Cleaved-Caspase-3 (1:100; Cell Signaling), GFAP (1:500, Abcam) and NeuN (1:500, Abcam, Cambridge, UK). Brains were evaluated for tumor growth, signs of toxicity, and tissue damage.

## Statistical analysis

All statistical analyses were carried out using GraphPad Prism® (Version 8.1, GraphPad Software, San Diego, CA). One-way ANOVA with Bonferroni or Tukey post-tests or a nonparametric Kruskal-Wallis test were performed, depending on the data distribution.

The *in vitro* inhibitory concentration 50% (IC$_{50}$) is reported as the mean of six independent experiments for each line, plotted relative to viable vehicle-normalized cells. Overall survival (OS) was calculated from the time of xenograft implant to death from tumor growth/infiltration. Kaplan-Meier survival curves with significance levels determined with log-rank test, and P values <0.05 were used for statistical significance. For *in vivo* studies, the Wilcoxon-Mann-Whitney test was used to determine a sample size of n = 6 and 8 per group, based on 80% power (5% significance; two-sided difference of means) and a standardized effect-size (signal/noise ratio of 1.6) estimated with preliminary studies on D425med-implanted mice.

## Results

### *In vitro* cytotoxic, anti-proliferative, and anti-clonogenic activity of DSF-Cu$^{++}$

DSF has recently shown proof of cytotoxic activity against a range of tumor histotypes. Here, we tested its anti-cancer effects on five pediatric MB cell lines. ONS76 (Shh-driven lines with wild-type Tp53), UW228 (Shh-driven lines with mutated Tp53), D425med, D341 and D283 (Group 3) cells were treated with an equimolar combination of DSF-Cu$^{++}$. On Cell-Counting-Kit-8 (CCK-8) assays, IC$_{50}$ ranged between 70 and 250nM after 48h, with an IC$_{95}$ of approximately 500nM in all tested lines, confirming DSF/Cu$^{++}$ cytotoxicity *in vitro* (Fig 1A–1E). Cu$^{++}$ alone was not toxic in MB lines at concentrations ranging from 500nM to 4μM, while DSF could not achieve its full cytotoxic potential without the co-administration of Cu$^{++}$ (S1 Fig).

We then turned to investigating the anti-proliferative and cell-cycle changes following treatment with DSF and combination DSF-Cu$^{++}$. Both treatments showed a significant anti-proliferative effect when cells were exposed continuously for a period of 144 hours, significantly slowing cell division at a concentration of 150nM (S1 Fig). Similarly to what was observed in the CCK-8 assay, the anti-proliferative potential of DSF was fully achieved only after co-dosing of Cu$^{++}$ in all the tested lines. Interestingly, the clonogenic potential of UW228 and ONS76 cells was also significantly impaired by DSF-Cu$^{++}$, with minimal observable colony growth at 150nM after an incubation of 7 days (S2 Fig).

### DSF/Cu$^{++}$ induces apoptosis and an increase in sub-G$_0$/G$_1$ cells on cell cycle analysis

We next investigated the effect of DSF-Cu$^{++}$ on cell apoptosis/necrosis at different concentrations and time points. The fractions of single-positive (AnnV$^+$-PI$^-$) and double-positive (AnnV$^+$-PI$^+$) cells were quantified with flow cytometry, and found to be significantly larger after 12-24-48h of treatment with 150nM DSF-Cu$^{++}$. Specifically, ONS76, UW228, D425 and D283 cells showed a significant increase in early apoptotic (AnnV$^+$-PI$^-$ cells), while all lines had higher levels of necrotic cells (AnnV$^+$-PI$^+$) at 24/48h post-treatment compared to control (Fig 1F–1J; S3 Fig).

Cell cycle arrest was quantified by Ki67 expression and DNA content in all cell lines. Ki67$^+$ cells were significantly decreased after 24h and 48h of treatment with DSF-Cu$^{++}$ at an equimolar concentration of 150nM and 300nM, and an increase in sub-G$_0$/G$_1$ cells was detected after 48h of treatment via PI staining in all tested cell lines (S4 **and** S5 **Figs**).

### DSF/Cu$^{++}$ reduces ALDH activity, CD133 expression, and neurosphere formation

As a number of studies in Atypical Theratoid/Rabdoid Tumor (AT/RT) and breast cancer have shown, DSF with the addition of Cu$^{++}$ reduced the expression of cancer stem-like cells (CSCs) by decreasing levels of ALDH, CD133 and Nestin, three markers of CSCs in brain malignancies [13, 19]. However, recent evidence in non-CNS malignancies pointed in the opposite direction, suggesting that ALDH inhibition takes place only after membrane permeabilization and cell death. In our studies, we first investigated neurosphere formation, that was found to be significantly impacted after 72 hours of treatment with equimolar concentrations of 150nM DSF-Cu$^{++}$ in D425med and D341 cells (S6 Fig **Panel A, B**). ALDH activity was then quantified by flow cytometry, and ALDEFLUOR$^®$ staining of D425med and D341 cells demonstrated a significant decrease in ALDH$^+$ cells compared to controls only in D341 cells, while ALDH activity was not suppressed by DSF and Cu$^{++}$ alone (Fig 2B) and by DSF-Cu$^{++}$

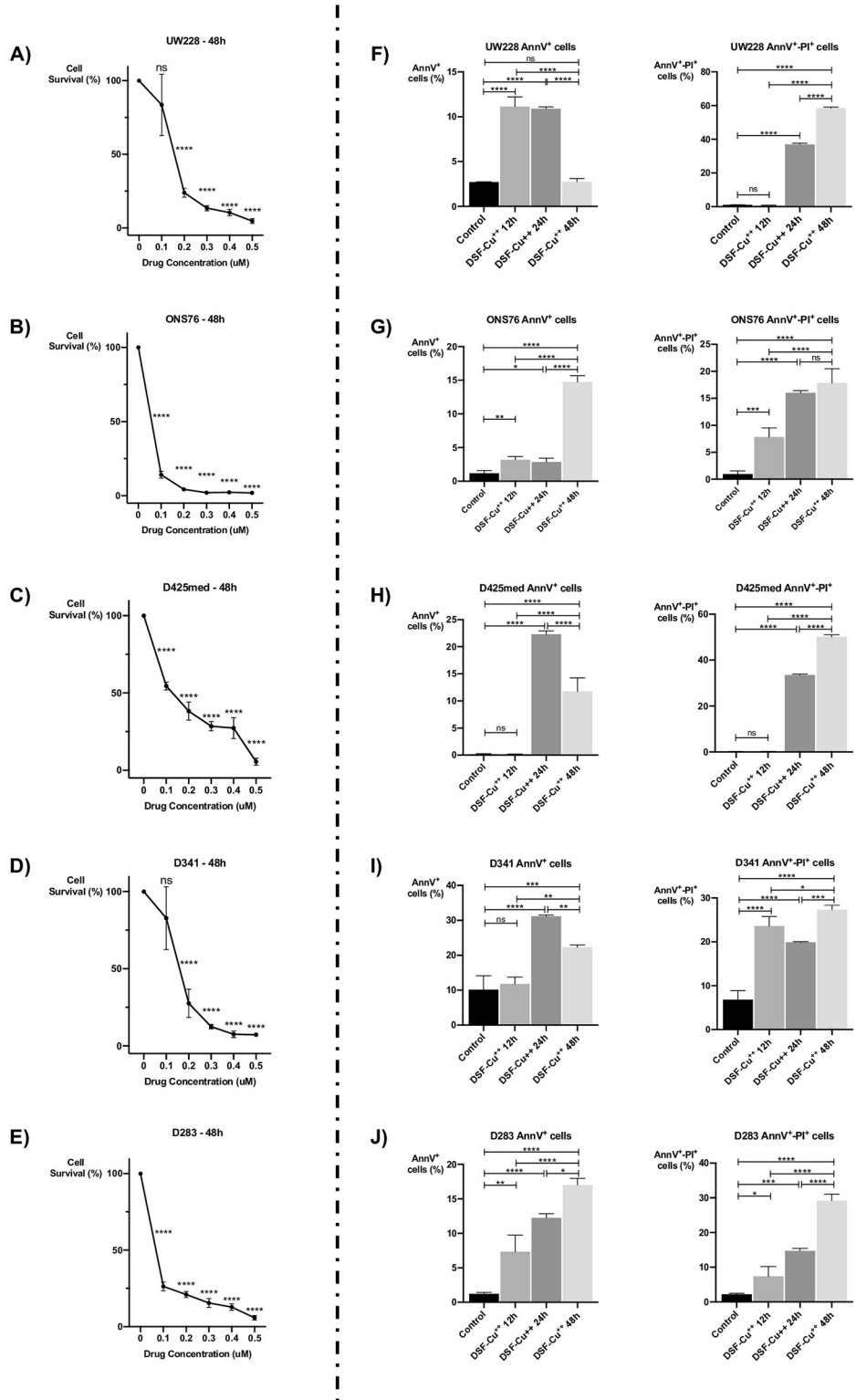

**Fig 1. DSF-Cu++ induces cells death and apoptosis in Shh-driven and Group 3 medulloblastoma. (A-B-C-D-E)** cytotoxicity assays with different concentrations of DSF-Cu++ in ONS76, UW228, D341, D425med and D283 cells. IC50s were between 100nM and 200nM in all the lines. **(F-G-H-I-J)** assessment of early and late apoptotic cells with AnnV-PI in the same cell lines. A significant increase in AnnV+-PI- cells was found at 12-24h post-treatment, while a larger fraction of AnnV+-PI+ cells was noticed after 24-48h of treatment with an equimolar dose of 150nM DSF-Cu++.

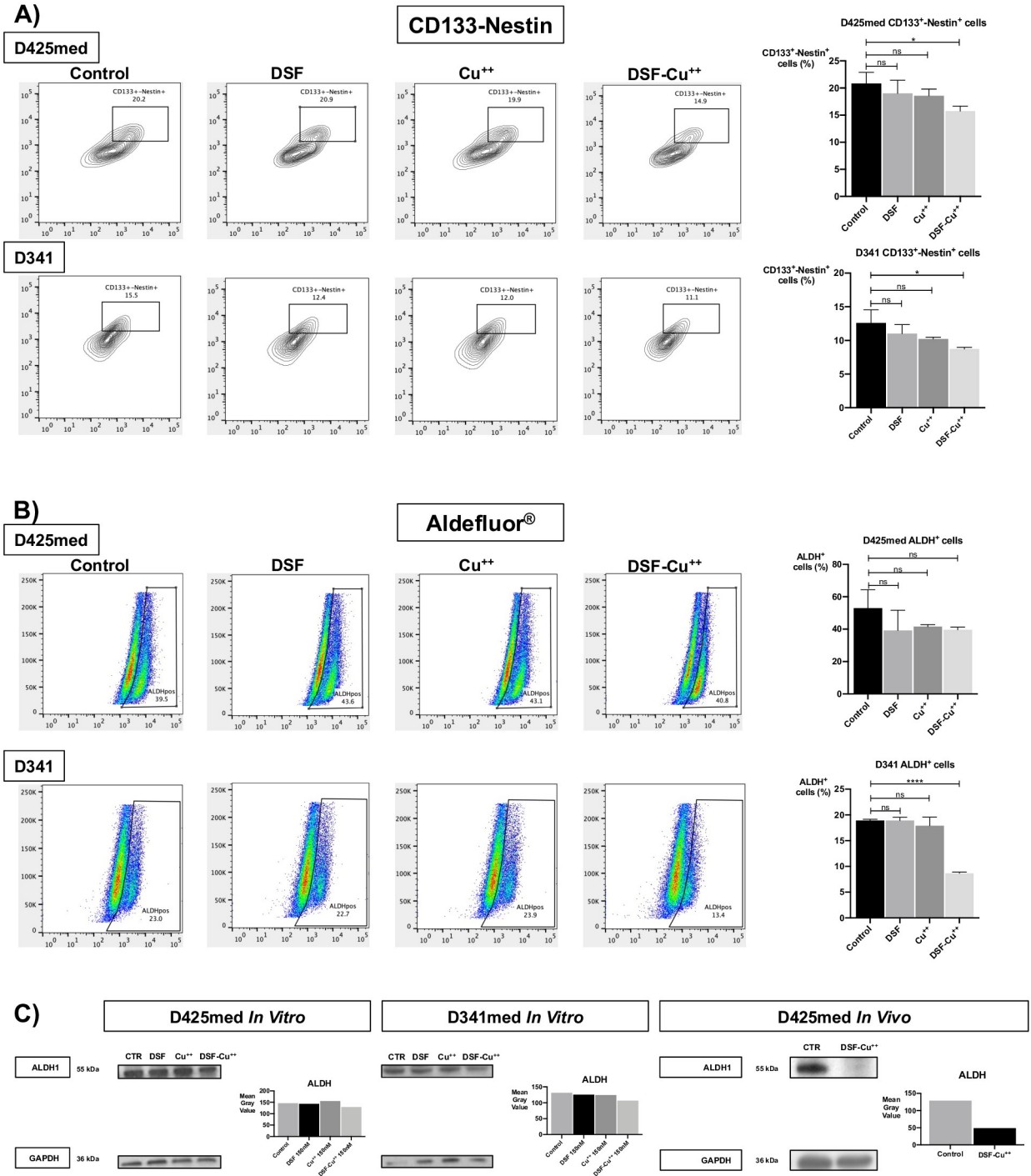

**Fig 2. DSF-Cu$^{++}$ reduces CD133$^{+}$-Nestin$^{+}$ and ALDH$^{+}$ cancer cells in D425med and D341 lines. (Panel A)** CD133$^{+}$-Nestin$^{+}$ D425med and D341 cells were significantly reduced by treatment with DSF-Cu$^{++}$ 150nM for 12h, while no significant effect was noticed in cells treated with DSF and Cu$^{++}$ only. Similarly, a reduction in ALDH, a key enzyme in stem-cells maintenance, was detected in DSF-Cu$^{++}$-treated cells, with minimal effect of DSF and Cu$^{++}$ only **(Panel B)**. Western blotting confirmed a significant reduction in ALDH expression after combination therapy, both *in vitro* and *in vivo* **(Panel C)**.

combination in D425med cells. Interestingly, a significant decrease in ALDH+ cells was found when higher concentrations of these drugs were used (S6 Fig **Panel C**) and after *in vitro* treatment of D425med and D341 cells for 12 hours (WB analysis in Fig 2C). Finally, *ex vivo* WB analysis on DSF-Cu++-treated D425med tumors showed a significant decrease in ALDH expression compared to untreated controls (Fig 2C).

In addition, DSF-Cu++ effect on CSCs was investigated with flow cytometry analysis of other common surface markers, such as CD133 and Nestin. In particular, D425med and D341 MB lines showed a significant reduction in CD133+/Nestin+ cells when compared to DSF, Cu++ and control after low concentrations of DSF/Cu++ for 12 hours, suggesting and early, despite moderate, anti-CSC effect (Fig 2A).

## Molecular mechanisms of DSF-Cu++ therapeutic effects in medulloblastoma

Skrott et al. have recently suggested NPL4, a key component of the p97 segregase cascade, as the main target modulating the chemotherapeutic effect of DSF-Cu++ [20]. The inhibition of the p97 pathway is thought to be responsible for an unfolded protein response (UPR) and the accumulation of intracytoplasmic and nuclear foci of poly-ubiquitylated proteins, with development of a Heat-Shock Response (HSR) and cell death.

NPL4-enrichment was therefore investigated in four lines with WB with multiple methodologies (UW228, ONS76, D425med, D341), and an increased expression was detected in cells treated for 24h with concentrations of 500nM and 1μM. Similarly, immunofluorescence analysis with a concentration of 150nM for 24h confirmed NPL4 nuclear sequestration and clustering, compared to controls (**Figs** 3A **and** 4; S7 Fig; S8 Fig **Panel A**; S9 Fig **Panel D**).

The Transcription factor 11 (TCF11) or Nuclear respiratory factor 1 (NRF-1), an endoplasmic reticulum-tethered leucine-zipper transcription factor activated from the 120 kDa into a 110 kDa isoform by cleavage, is also dependent on the p97 segregase cascade [20]. Interestingly, significant accumulation of the 120 kDa isoform and additional cleavage into several other variants with a range of different MWs was noticed after treatment, suggesting the involvement of the p97 pathway in the mechanism of action of DSF-Cu++ (Fig 3A and 3B; S8 Fig **Panel A and B**, S9 Fig **Panel D**). Similarly, treatment of D341 and D425med cells with ML240, a strong p97-inhibitor, showed clustering of NRF-1 fragments around the 120 kDa mark, suggesting that this pathway might be the shared mechanism of action between ML240 and DSF-Cu++. Finally, dose-dependent accumulation of poly-ubiquitylated proteins was detected in cells treated with a range of DSF-Cu++ concentrations, a downstream effect of p97 blockade that was shown to lead to HSR and cell death, and common to other non-CNS tumors [20] (Fig 3C).

## DSF-Cu++ impairs mechanisms of DNA repair

DNA damage and γ-H2AX, the phosphorylated variant of histone H2A that arises after a genotoxic stress, have been suggested as potential mediators of DSF-Cu++ cytotoxicity in glioblastoma [14]. In medulloblastoma, WB and immunofluorescence analysis of γ-H2AX expression revealed increased protein expression *in vitro* and in *ex vivo* WB, and a higher number of γ-H2AX+ nuclear foci after 48h of treatment (Fig 3 **Panels A-D**; Fig 4; S7 Fig; S8 Fig **Panel A**; S9 Fig **Panel D**), suggesting that the combination of DSF-Cu++ might induce DNA damage in tumor lines and *in vivo*. To further support these findings, we measured the cellular levels of Checkpoint kinase 1, a mediator that regulates DNA-damage response and the G2/M phase transition checkpoint, and of its active counterpart, Phospho-Chk-1 (Phospho-Checkpoint kinase 1) [14]. *In vitro* WB analysis on MB lines showed a moderate decrease in the expression

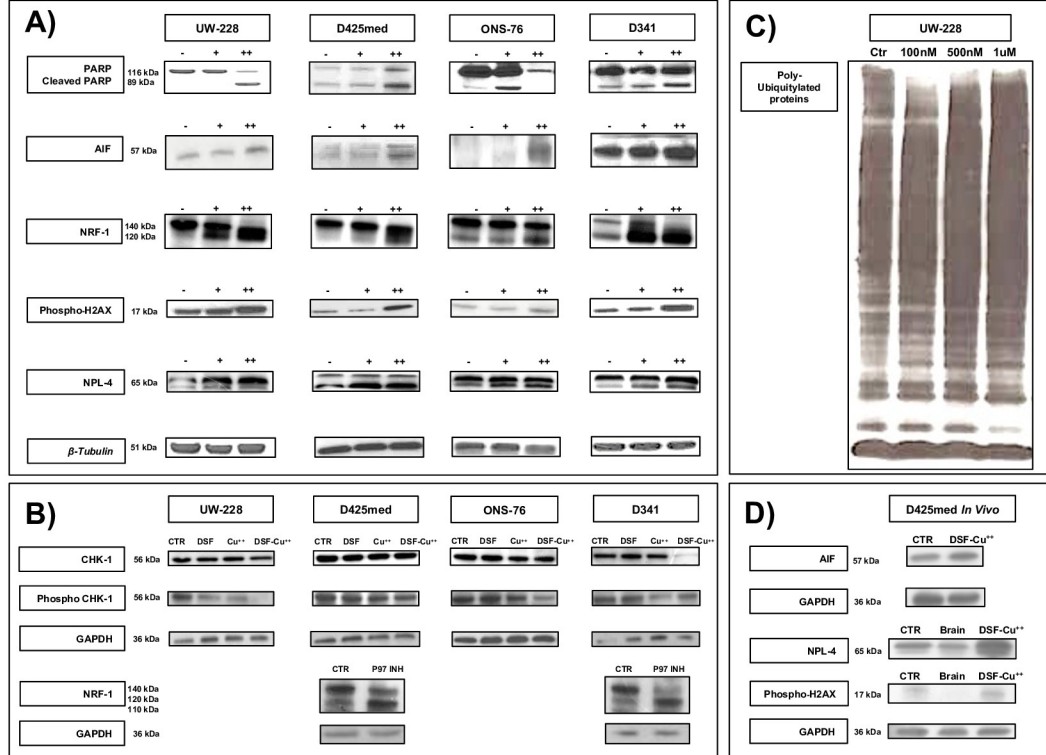

**Fig 3. Western blot of MB lines treated with DSF/Cu$^{++}$.** (**Panel A**) Cleaved-PARP fraction and AIF increased after DSF-Cu$^{++}$ treatment, as well as NPL4 and Phospho-H2AX. Multiple fractions of NRF1 confirmed the inhibition of this transcription factor, and a similar cleavage profile was found after treatment of D425-D341 cells with a p97-inhibitor, suggesting the involvement of this intracellular pathway (**Panel B**). Changes in Chk-1 and Phospho-Chk-1 suggest an effect of DSF-Cu$^{++}$ on the DNA repair mechanisms, and a reduction in these markers is triggered only by the use of combination DSF-Cu$^{++}$ (**Panel B**). Furthermore, DSF-Cu$^{++}$ and CUET consistently determine an intracytoplasmic accumulation poly-ubiquitylated proteins in UW228 cells (**Panel C**). *In vivo* western blotting of control and DSF-Cu$^{++}$-treated tumor tissue, as well as of normal brain, showed an increase in AIF, NPL4 and H2AX after treatment, consistently with *in vitro* findings (**Panel D**). Legend;—control, + 0.5μM for 24h; ++ 1μM for 24h; DSF 150nM 24h, Cu$^{++}$ 150nM 24h, DSF-Cu$^{++}$ 150nM 24h. *In vivo* DSF-Cu$^{++}$-treated brains were obtained from D425med tumors treated with oral DSF (150mg/kg daily) and Cu$^{++}$ (2mg/kg daily).

of these markers after exposure to DSF-Cu$^{++}$, corroborating the hypothesis that DSF-Cu$^{++}$ cytotoxicity could partly be caused by DNA-damage response disruption [14] (Fig 3B; S8 Fig **Panel B**).

## DSF-Cu$^{++}$ induces apoptosis and cell death through the release of Apoptosis-Inducing Factor (AIF) and poly(ADP-ribose)polymerase (PARP) activation

Given the known cytotoxic effects of DSF-Cu$^{++}$, we then turned our attention to the understanding of the molecular mechanisms responsible for apoptosis induction observed in medulloblastoma. In MB lines, WB analysis showed a uniform increase in the cleaved-PARP fraction and reduction of total PARP after 24h of treatment with 500nM and 1μM of DSF/Cu$^{++}$ (Fig 3 **Panel A**; S8 Fig **Panel A**). Similarly, AIF levels, a caspase-independent effector that initiates nuclear DNA condensation and regulates mitochondrial permeability, were found to be increased on *in vitro* and *ex vivo* immunoblotting on brains treated with DSF-Cu$^{++}$ (Fig 3

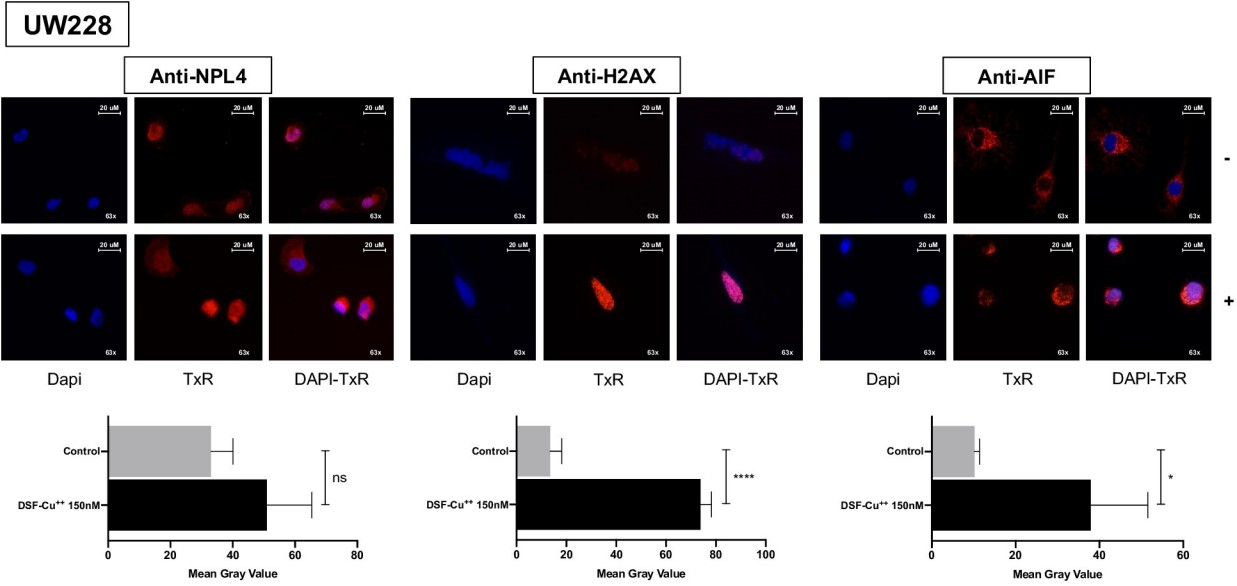

**Fig 4. DSF-Cu++ induces nuclear translocation of AIF, and clustering of H2AX and NPL4. A)** Higher NPL-4 clustering in the cell cytoplasm and nucleus was noticed after treatment with DSF-Cu++, compared to control. **B)** nuclear foci of H2AX appear in the nuclei of cells treated for 24h with 150nM DSF-Cu++. **C)** UW228 cells showed nuclear translocation of AIF, a marker of apoptosis, after DSF-Cu++ treatment. Alternatively, this marker accumulates in the cytoplasm in control cells. Images were acquired at 63x magnification. Legend;—control, + 150 nM for 24h.

Panel A and D; S8 Fig **Panel A and D**). AIF translocation was also detected via immunofluorescence in apoptotic cells, confirming the movement from the cytoplasm to the nucleus–signaling the initiation of the apoptotic cascade—in UW228/ONS76/D524/D341 cells treated with DSF-Cu++ for 24h (Fig 4A **and** S7 **and** S8 **Figs Panel A**, S9 Fig **Panel D**).

## DSF-Cu++ inhibits tumor growth and prolongs survival *in vivo*

As the *in vitro* data indicated that DSF-Cu++ induces apoptosis in a wide range of Shh-driven and Group 3 lines at nanomolar concentrations, we decided to assess its anti-cancer potential and clinical applicability in well-established intracranial models of pediatric MB. First, we investigated the safety profile of orally administered DSF-Cu++, and showed that three daily doses of DSF (50/150/250 mg/kg) combined with Cu++ (2 mg/kg) did not result in overt toxicity and changes in animal behavior. Moreover, animal weight remained stable, with normal increase, during the three weeks of the study, except for occasional signs of local dermatitis and skin irritation (Fig 5C; S10A Fig).

We then focused on the therapeutic efficacy of DSF-Cu++ in two intracranial murine models of MB. To evaluate the efficacy of this combination in the most realistic and challenging setting, we utilized two aggressive Group 3 cell lines derived from pediatric patients with and without spinal "drop" metastases. After cortical implantation of $1.5 \times 10^5$ cells, mice were treated twice daily with oral DSF (at a dose of 150 mg/kg/day), and once a day with oral Cu++ (2 mg/kg/day) (Fig 5A and 5B **and** S10B Fig). To better reproduce the clinical scenario we chose to administer Antabuse®, the trade name of DSF used in patients with chronic alcoholism [23, 24], and Cu++ bisglycinate, a common over-the-counter Cu++ supplement. Animals were assessed twice daily for neurologic symptoms and signs of cachexia, and sacrificed when ACUC criteria were met. Survival was significantly prolonged in the treatment group, compared to controls, in both lines. In particular, in animals implanted with D425med, a median

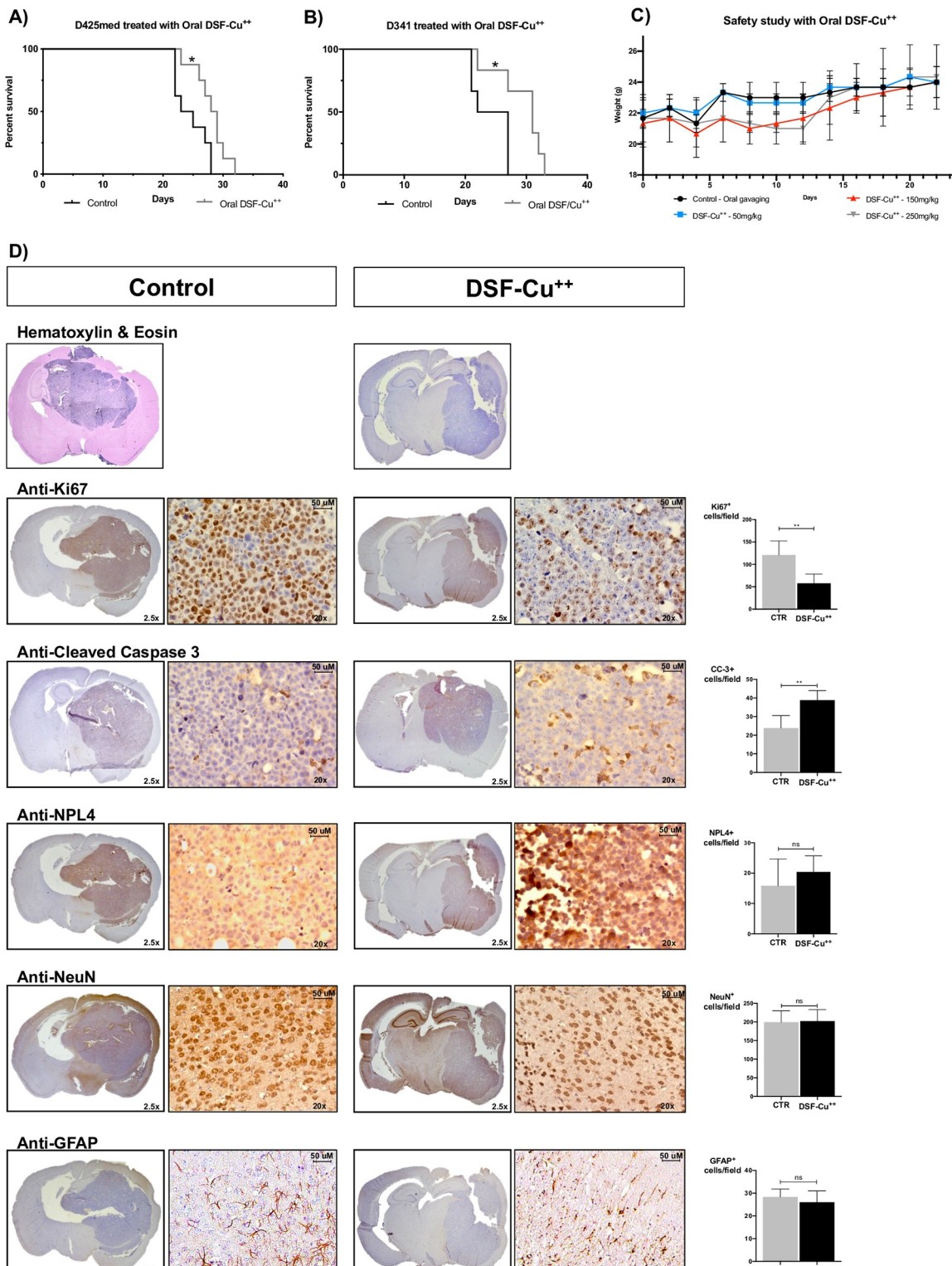

**Fig 5. Kaplan-Meier curve, H&E, and IHC analyses. A-B)** Survival was assessed in a D425med and D341 model of MB, with intracranial implantation of Group 3 MB cells and daily treatment with DSF-Cu++. In D425med-implanted animals, a median survival of 28.5 days was achieved in treated animals, compared to 24 days in controls (P = 0.02). In mice bearing D341 tumors a median survival of 31 days was achieved in the treatment group, with controls reaching 24.5 days. **C)** Safety study in Nu/Nu Athymic mice treated with oral gavage of water, DSF (50/150/250mg/kg) and Cu++ (2mg/kg). **D)** Histopathological analysis of control and DSF-Cu++-treated brains. H&E showed

the morphology and implantation site of D425med tumors, while IHC confirmed a higher expression of Ki67+ cells in control brains, an increase in Cleaved-Caspase-3+ cells in DSF-Cu++-treated animals and NPL4 clustering in the cytoplasm and nucleus of cells treated with this combination. GFAP and NeuN staining showed similar number of astrocytes and neurons in control and treated animals. Images were acquired at 2.5x and 20x magnification.

survival of 28.5 days was achieved in animals treated with DSF/Cu++, 4.5 days longer than that of untreated animals (24 days; p = 0.02; HR 0.21; 95% CI 0.06–0.78) (Fig 5A). Similarly, mice bearing D341 tumors reached a median survival of 24.5 days when untreated, compared to 31 days in the DSF-Cu++ group (HR 0.14; 95% CI 0.02–0.77) (Fig 5B). These results suggest that a combination of DSF and Cu++, alone or in association with other approaches, might be explored clinically in the future to improve survival outcomes in patients with medulloblastoma.

## DSF-Cu++ enhances tumor cell apoptosis, NPL4 expression and reduces tumor proliferation *in vivo*

To confirm the survival data and support them with an independent measure of tumor burden, all the brains were extracted, fixed, paraffin-embedded, and assessed histologically for tumor burden. Significantly higher levels of Cleaved-Caspase-3 expression were found in tumors treated with DSF-Cu++ on immunohistochemical analysis, demonstrating treatment-induced tumor cell death (Fig 5D). A significant decrease in Ki67 expression was also noticed in animals that received the combination treatment, a finding consistent with WB and immunofluorescence assays at nanomolar concentrations of DSF-Cu++ (Fig 5D). Finally, higher intranuclear expression of NPL4 in treated brains compared to untreated controls was detected, both with *ex vivo* WB and IHC (**Figs** 3D **and** 5D; S9 Fig **Panel D**). No signs of neuronal or astrocytic toxicity were reported after staining for NeuN and GFAP, respectively (Fig 5D).

## Discussion

Despite having seen a remarkable improvement in patient survival over the last few decades, medulloblastoma (MB) still remains a significant clinical and research challenge for multiple reasons. The current standard of treatment involves the use of maximal safe surgical resection, radiation and chemotherapy with a multi-agent regimen, resulting in a 5-year overall survival (OS) of 50–55% in high-risk Groups (p53-mutated Shh-driven and Group 3). However, prognosis is still poor in children under 3 years of age and in those presenting with Tp53-mutated Shh-driven and Group 3 MBs. These molecular subgroups of MB are characterized by a particularly aggressive tumor growth, with a five-year OS of 41% and 50%, respectively, frequent drop metastases to the leptomeningeal spaces, and high-level expression and amplification of MYC [1, 2, 25–29]. Novel therapeutic approaches and compounds are therefore urgently needed to improve patient survival and quality of life, and to reduce the severity of treatment-related toxicities. Unfortunately, the development of anti-cancer agents is a long and expensive process, and only a small number of new molecules reach the market [30]. Alternatively, repurposing old compounds already approved by the FDA for non-cancer indications is a safe and effective option guided by available phase III/IV data, which streamlines their evaluation for additional indications [31].

Disulfiram, a molecule used for decades in the treatment of chronic alcoholism (trade name: Antabuse®), has recently been repurposed as an anti-cancer compound in several malignancies, including multiple Phase I, II and III clinical trials in patients with recurrent

glioblastoma [23, 24, 32]. Interestingly, its cytotoxic potential is further enhanced by the co-administration of Cu$^{++}$, and the resulting combination achieved a dramatically lower IC$_{50}$ in several tumors of different origin [33]. DSF, after being absorbed from the gastrointestinal tract, reacts with Cu$^{++}$ to form diethyldithiocarbamate (DDC), diethyldithiomethylcarbamate (Me-DDC), diethylthi-omethylcarbamate (Me-DTC), and other metabolites active against the liver enzyme ALDH [23, 34, 35]. The addition of divalent ions such as copper triggers the synthesis of bis(diethyldithiocarbamate)-copper (CuET), a metabolite responsible for the *in vivo* therapeutic effect of DSF-Cu$^{++}$. CuET readily crosses the BBB, and is responsible for the cytotoxic activity of DSF-Cu$^{++}$ in cancer (Fig 6) [20, 22].

In this study we aimed at characterizing the main targets of DSF-Cu$^{++}$ in medulloblastoma, introducing a novel repurposed chemotherapeutic agent for medulloblastoma, and also providing a solid base for future exploration of its clinical uses. The uncovering of these mechanisms will facilitate the repositioning of DSF-Cu$^{++}$ in pediatric MB, and will pave the way for the introduction of new compounds and strategies to treat of this aggressive brain tumor.

## Molecular mechanisms of DSF-Cu$^{++}$ toxicity in medulloblastoma

In the present study, we provide evidence of the anti-tumor effect of DSF/Cu$^{++}$ and explain their mechanisms of action *in vitro* and *in vivo*. Based on the data obtained by our group and that provided by other investigator in different tumors, their cytotoxic effect is likely secondary to the intracellular accumulation ubiquitylated proteins, with consequent alteration of the physiologic pathways of protein disposal and degradation, and appears to be shared among CNS and non-CNS tumors. By inducing NPL4 aggregation and intracellular—both cytoplasmic and nuclear—clustering, DSF and CuET contribute to the inhibition of the p97 ATPase cascade, a pathway responsible for the extraction of poly-ubiquitylated (poly-Ub) proteins from cellular structures such as the endoplasmic reticulum and Golgi apparatus, with subsequent proteasomal degradation (Fig 6) [20, 22]. Furthermore, through NRF1 blockade, a transcription factor related to the p97 cascade, CuET is believed to promote the cytoplasmic accumulation of poly-ubiquitylated proteins, with apoptosis quickly following treatment and elevation of both cleaved-PARP and Apoptosis Inducing Factor (AIF) fractions. Similar mechanisms of apoptosis and cell-cycle arrest were reported across all cell lines tested. Interestingly, the tumor origin and molecular subgroup did not seem to play a role in DSF-Cu$^{++}$ sensitivity in medulloblastoma, a finding that supports their use in these aggressive histotypes.

## DSF-Cu$^{++}$-mediated impairment of DNA repair mechanisms

Interestingly, another mechanism responsible for the cytotoxic effect of these drugs appears to involve DNA and histone damage. Previous studies have suggested the involvement of the DNA damage repair system as one of the main targets of DSF-Cu$^{++}$ toxicity, with higher expression of H2AX correlating with histone phosphorylation and DNA alteration [21]. The alteration of DNA damage repair mechanisms might also explain the combinatory and radio-sensitizing effects of DSF-Cu$^{++}$ and Temozolomide-Radiation found in other cancers, such as glioblastoma and AT/RT[14, 21]. Furthermore, the inhibition of a Checkpoint-kinase (Chk1 and Phospho-Chk1), also involved in DNA repair, reinforces the idea that the combination of DSF-Cu$^{++}$ might exert part of its cytotoxic potential via DNA damage induction in MB.

## DSF-Cu$^{++}$ effect on cancer stem cells markers

Cancer stem cells (CSCs) are a rare population responsible for tumor initiation, recurrence and metastasis. Commonly defined as brain tumor stem cells (BTSCs) or initiating cells (BTICs), CSCs express specific enzymes and surface markers that allow their identification by

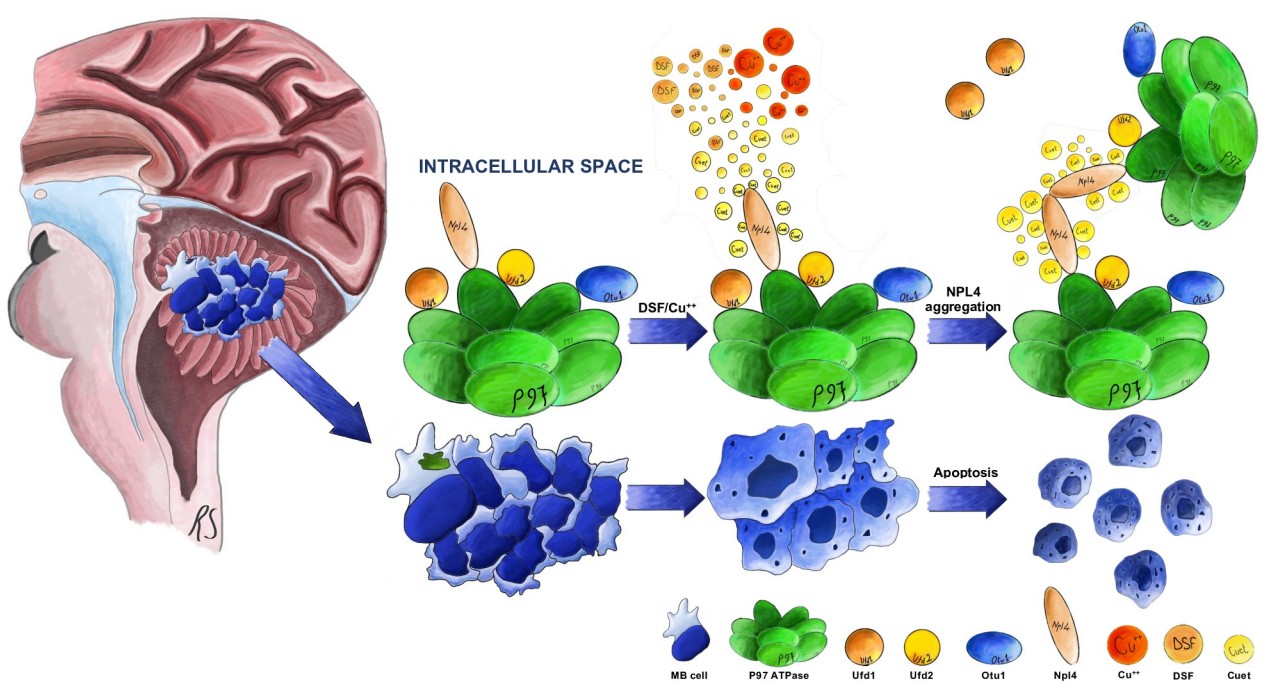

**Fig 6. Molecular targets of DSF-Cu++.** Schematic of the molecular mechanisms of action of DSF-Cu++.

means of WB, immunofluorescence and FACS analysis [36, 37]. In particular, brain tumors have high levels of ALDH, CD133, and Nestin, with ALDH+ and CD133+ cells accounting for about 8% and 17–40%, respectively, in pediatric MB [38–41]. In this study we assessed and demonstrated that this combination was not only capable of killing MB cells, but that CSCs were specifically targeted and eliminated. DSF-Cu++ proved effective at reducing the tumor-initiating fraction at nano- and micromolar concentrations after only hours of direct exposure. These data, supported by a significantly lower neurosphere count in serum-free media, provide extensive evidence for the possible use of this combination as a cytotoxic and anti-CSC agent in patients. The effects of DSF-Cu++ on CSCs would in fact help targeting with higher precision those tumors that express a large fraction of ALDH+, CD133+ and Nestin+ cells, which might otherwise be more prone to treatment resistance, drop-metastasis and recurrence.

In consideration of the significant cytotoxicity, anti-stem cell-ness, and anti-proliferative effects of DSF-Cu++ *in vitro*, we hypothesize that the optimization of their delivery across the BBB and throughout the CNS may achieve even better results *in vivo*, broadening the choice of therapeutic strategies for this particularly challenging tumor. To address this, we tested the combination of DSF-Cu++ on animal survival in two xenograft models of MB. A significant survival increase in treated animals was shown, supporting the hypothesis that DSF-Cu++ can prolong survival without causing serious toxicities or side effects.

As with all studies, ours includes limitations. Among them was the tumor models, not reproducing completely the clinical characteristics and behavior of pediatric MB, as well as the timing and dosage of drug administration that would need to be tested in a clinical setting. The limited increase in median survival could be due to the low concentrations of DSF-Cu++ achieved in the brain and within the tumor mass, an issue common to a number of chemotherapeutics used in brain malignancies. Biodegradable polymers, nanoparticles, or convection-enhanced delivery might confer a better distribution of DSF-Cu++ in the CNS and lead to

more significant results than systemic administration. The integration of these strategies could provide a valid alternative to those patients not responding to standard regimens, shedding light on the use and translatability of DSF, giving new life to this old and safe compound, and paving the way for repurposing other drugs in the future. Therefore, additional studies are needed to better understand the role of DSF-Cu$^{++}$ as a chemotherapeutic option for pediatric MB. A recent Phase I/II study of DSF and Cu$^{++}$ with concurrent radiation therapy and TMZ for newly diagnosed glioblastoma demonstrated promising preliminary efficacy for a subset of tumors with *IDH1*, *BRAF*, and *NF1* mutations [32]. These data support the use of this combination in a subset of patients, and the possibility of expanding the current trials to include larger samples of newly diagnosed glioblastoma. Similarly, the identification of specific subsets of medulloblastoma might help selecting those patients that would benefit the most from this combination therapy.

In conclusion, we demonstrated that the combination of DSF and Cu$^{++}$ is able to achieve significant *in vitro* cytotoxicity at nanomolar levels, as well as suppression of cancer cell stemness, cell cycle blockage, and reduction of surface and nuclear markers of proliferation. Furthermore, we showed efficacy of DSF-Cu$^{++}$ *in vivo* in two separate models of pediatric MB, with a significant increase in animal survival and an overall positive safety profile, providing evidence on the possible future use of these drugs in combination with radio- and chemotherapeutic regimens.

## Supporting information

**S1 Fig. Proliferation assay.** Combination of DSF and Cu$^{++}$ determines a significant reduction in proliferative rate in all the lines when compared to untreated controls and DSF-treated cells. Cells were treated after 24h from plating with DSF or DSF-Cu$^{++}$ 150nM and counted at 96h and 144h post-treatment.
(PDF)

**S2 Fig. Clonogenic assay.** ONS76 and UW228 clonogenic potential is significantly reduced by 7 days of combination therapy with DSF-Cu$^{++}$ 150nM.
(PDF)

**S3 Fig. Flow cytometry analysis of early and late apoptotic cells.** AnnV/PI staining showed a significant increase in AnnV$^+$-PI$^-$ and AnnV$^+$-PI$^+$ cells after 12-24h and 24-48h, respectively. DSF-Cu$^{++}$ was used at a concentration of 150nM in all cell lines. SSC/FSC gating is shown for each cell line, as well as AnnV/PI gating in control and DSF-Cu$^{++}$-treated samples.
(PDF)

**S4 Fig. DSF-Cu$^{++}$ reduces Ki67$^+$ cells in UW228, ONS76, D425med, D341 and D283 cells.** SSC/FSC and Ki67 gating is shown for all five cell lines (control and DSF-Cu$^{++}$-treated 150 nM for 24h). Statistical analysis of cells treated with DSF-Cu$^{++}$ 150nM and 300nM for 24h and 48h is shown on the right.
(PDF)

**S5 Fig. Cell cycle analysis.** Cell cycle analysis through PI staining of DNA content was carried out in ONS76, UW228, D425med and D341 cell-lines. An increase in sub-G0/G1 cells was noticed after 48 hours of treatment with 150nM DSF-Cu$^{++}$, consistently with AnnV/PI data.
(PDF)

**S6 Fig. Neurosphere formation assay.** Neurosphere formation was greatly reduced after 72 hours of treatment with 150nM DSF-Cu$^{++}$ in D425med and D341 cells. **High-Dose DSF-Cu$^{++}$**

**reduces ALDH$^+$ cancer cells in D425med and D341 lines**. 2uM DSF-Cu$^{++}$ induces significant reduction in ALDH$^+$ cells in D425med and D341 after 2 hours of treatment.
(PDF)

**S7 Fig. Combination DSF-Cu$^{++}$ induces nuclear translocation of AIF, and clustering of H2AX and NPL4 in all lines.** A) ONS76, D425med and D341 cells showed nuclear and cytoplasmic clustering of NPL4, expression of nuclear foci of H2AX and AIF translocation after treatment with DSF-Cu$^{++}$. Signal intensity was analyzed with ImageJ and plotted with GraphPad Prism on three replicates per group. Legend;—control, + 150 nM for 24h.
(PDF)

**S8 Fig. Band intensity quantification of western blots presented in Fig 3—*In vitro blots.*** Western blot films presented in Fig 3 obtained from *in vitro* assays were quantified with ImageJ for band intensity and plotted with GraphPad Prism.
(PDF)

**S9 Fig. Band intensity quantification of western blots presented in Fig 3—*In vivo blots.*** Western blot films presented in Fig 3 obtained from *in vivo* experiments were quantified with ImageJ for band intensity and plotted with GraphPad Prism.
(PDF)

**S10 Fig. Safety study in healthy Nu/Nu athymic mice and dosing schedule for safety and efficacy studies presented in Fig 5.**
(PDF)

## Acknowledgments

We thank the Hunterian Neurosurgical Research Laboratory for the support and guidance throughout the entire project.

## Author Contributions

**Conceptualization:** Riccardo Serra, Noah Leviton Gorelick, Joshua Casaos, Arba Cecia, Antonella Mangraviti, Alessandro Olivi, Henry Brem, Eric M. Jackson, Betty Tyler.

**Data curation:** Riccardo Serra, Antonella Mangraviti, Charles Eberhart, Betty Tyler.

**Formal analysis:** Riccardo Serra, Tianna Zhao, Noah Leviton Gorelick, Henry Brem.

**Investigation:** Riccardo Serra, Tianna Zhao, Sakibul Huq, Noah Leviton Gorelick, Joshua Casaos, Arba Cecia, Renyuan Bai, Eric M. Jackson.

**Methodology:** Riccardo Serra, Tianna Zhao, Sakibul Huq, Noah Leviton Gorelick, Joshua Casaos, Arba Cecia, Betty Tyler.

**Project administration:** Betty Tyler.

**Resources:** Charles Eberhart, Renyuan Bai, Henry Brem, Eric M. Jackson, Betty Tyler.

**Supervision:** Tianna Zhao, Antonella Mangraviti, Renyuan Bai, Alessandro Olivi, Henry Brem, Eric M. Jackson, Betty Tyler.

**Writing – original draft:** Riccardo Serra, Antonella Mangraviti, Alessandro Olivi, Betty Tyler.

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
