## [Decision Letter · Decision Letter 0]

8 Feb 2021

PONE-D-20-38226

Disulfiram and Copper Combination Therapy Targets NPL4, Cancer Stem Cells and Extends Survival in a Medulloblastoma Model

PLOS ONE

Dear Dr. Serra,

Thank you for submitting your manuscript to PLOS ONE. After careful consideration, we feel that it has merit but does not fully meet PLOS ONE’s publication criteria as it currently stands. Therefore, we invite you to submit a revised version of the manuscript that addresses the points raised during the review process.

We look forward to receiving your revised manuscript.

Kind regards,

Ilya Ulasov, Ph.D

Academic Editor

PLOS ONE

Journal Requirements:

2)  Please state the source of mice used in the study.

3) Please provide the product number and any lot numbers of the antibodies purchased for your study.

4) Thank you for stating the following in the Acknowledgments Section of your manuscript:

[COMPETING INTERESTS

Dr. Brem has research funding from NIH, Johns Hopkins University, Arbor Pharmaceuticals, Bristol-

Myers Squibb, and Acuity Bio Corp* and philanthropy. Dr. Brem is also a consultant for AsclepiX

Therapeutics, StemGen, InSightec, Accelerating Combination Therapies*, Camden Partners*, LikeMinds,

Inc*, Galen Robotics, Inc.* and Nurami Medical*. Betty Tyler is a consultant for Accelerating

Combination Therapies*. (*includes equity or options).

FUNDING

We would like to acknowledge Ms. Kimberly Spiro, the Donald W. Spiro Foundation, and Donald R.

Spiro whose generous funding made this work possible.]

 [The author(s) received no specific funding for this work.]

Additionally, because some of your funding information pertains to [commercial funding/COI], we ask you to provide an updated Competing Interests statement, declaring all sources of commercial funding.

In your Competing Interests statement, please confirm that your commercial funding does not alter your adherence to PLOS ONE Editorial policies and criteria by including the following statement: "This does not alter our adherence to PLOS ONE policies on sharing data and materials.” as detailed online in our guide for authors  http://journals.plos.org/plosone/s/competing-interests.  If this statement is not true and your adherence to PLOS policies on sharing data and materials is altered, please explain how.

Please include the updated Competing Interests Statement and Funding Statement in your cover letter. We will change the online submission form on your behalf.

5) PLOS ONE now requires that authors provide the original uncropped and unadjusted images underlying all blot or gel results reported in a submission’s figures or Supporting Information files. This policy and the journal’s other requirements for blot/gel reporting and figure preparation are described in detail at https://journals.plos.org/plosone/s/figures#loc-blot-and-gel-reporting-requirements and https://journals.plos.org/plosone/s/figures#loc-preparing-figures-from-image-files. When you submit your revised manuscript, please ensure that your figures adhere fully to these guidelines and provide the original underlying images for all blot or gel data reported in your submission. See the following link for instructions on providing the original image data: https://journals.plos.org/plosone/s/figures#loc-original-images-for-blots-and-gels.

6) Your ethics statement should only appear in the Methods section of your manuscript. If your ethics statement is written in any section besides the Methods, please delete it from any other section.

Reviewers' comments:

Reviewer's Responses to Questions

**Comments to the Author**

1. Is the manuscript technically sound, and do the data support the conclusions?

Reviewer #1: Yes

2. Has the statistical analysis been performed appropriately and rigorously? 

Reviewer #1: I Don't Know

3. Have the authors made all data underlying the findings in their manuscript fully available?

Reviewer #1: Yes

4. Is the manuscript presented in an intelligible fashion and written in standard English?

Reviewer #1: Yes

5. Review Comments to the Author

Reviewer #1: Chelation with copper is essential for the anticancer effects of the old alcohol reversion drug disulfiram. As such, administration of DSF along with copper gluconate has emerged as a drug repurposing strategy with clinical trials underway in multiple cancers. This strategy is of particular interest for brain cancers, because, DSF can cross the blood-brain barrier. There have been several preclinical studies published on DSF-Cu therapy in glioblastoma and other brain cancer models. The authors here have performed extensive cell culture and orthotopic xenograft studies using standard cytotoxicity, flow cytometry, western blot, and immunochemical analyses in a panel of human medulloblastoma cell lines. The results are along the expected lines of strong cytotoxicity, antitumor effects, expression of apoptotic regulatory proteins, etc by DSF-Cu. Apart from confirming the anticancer effects of Cu-DSF, this study lacks novelty and does not shed light on the probable mechanisms of action of the copper chelated disulfiram. However, it is acceptable for publication in PLOSone given the journal policy for acceptance. Nevertheless, the following points require attention before acceptance.

1) The strategy of copper gluconate supplementation with DSF has been tried in many trials, however, the results have not been encouraging. The authors should discuss the possible reasons and future directions in Cu-DSF treatments.

2) Sure Cu-DSF may cause aggregation of NPL4, however, this is not likely the only mechanism. Can NPL4 clustering be a response seen in dying cells? Were any major targets affected by NPL4 aggregation? Currently, there is no information on the role of NPL4 in therapy-targeted ubiquitin destruction of proteins. How were the cell extracts prepared for NPL4 Western blots? Soluble or insoluble fractions used for SDS-PAGE? If aggregated, much of the protein is likely to remain with cell debris after lysis. Did authors find NPL4 aggregation in xenografted tumor specimens after Cu-DSF administration?

3) How do the DSF and Cu doses used in animal experiments compare with those used in clinical trials?

6. PLOS authors have the option to publish the peer review history of their article (what does this mean?). If published, this will include your full peer review and any attached files.

Reviewer #1: No

---

## [Author Response · Author response to Decision Letter 0]

26 Apr 2021

Response to Reviewers 

We thank the Reviewer for her/his time and helpful comments on our manuscript. In response to 

her/his critiques, we have introduced several changes to our manuscript. Furthermore, in the present Letter we tried to explain our rationale and perspective on the points brought up by the Reviewer.

1) The strategy of copper gluconate supplementation with DSF has been tried in many trials, however, the results have not been encouraging. The authors should discuss the possible reasons and future directions in Cu-DSF treatments.

We thank the reviewer for this comment. Although some of the results of large randomized clinical trials in brain cancers with the combination of Disulfiram and Copper are not encouraging, a recent Phase I/II study of disulfiram and copper with concurrent radiation therapy and temozolomide for newly diagnosed glioblastoma showed promising preliminary efficacy for a subset of glioblastoma with IDH1, BRAF, and NF1 mutations. Among these patient glioblastomas with IDH1 (n = 6), BRAF (n = 2), or NF1 (n = 1) mutations had significantly better PFS and OS than those without the mutations (1-year PFS: 100% vs 22%, respectively, p = 0.001; 1-year OS: 100% vs 42%, respectively, p = 0.006) (2). These data might therefore support the use of this combination in specific subsets of patients, or at least the possibility of expanding the current trials to include larger samples of newly diagnosed glioblastoma. Furthermore, a major advantage of Disulfiram-Copper is their relatively benign and well-known safety profile, especially when compared to novel anti-cancer strategies. Finally, their limited cost will allow for a larger use and more widespread testability in resource-limited settings. As suggested by the Reviewer, we included these considerations in our Discussion.

2) Cu-DSF may cause aggregation of NPL4, however, this is not likely the only mechanism. Can NPL4 clustering be a response seen in dying cells? 

We thank the Reviewer for commenting on this important and still debated point of DSF cytotoxicity.

Skrott et al. showed that NPL-4 is usually not a physiologic response seen in cell death, and that it is triggered by the combination of DSF-Cu++ and their metabolite Cuet. Specifically, NPL-4 aggregation is the mechanism involved in p97 blockage, leading to the downstream activation of the Heat Shock Response that causes cell death(1).

However, we agree with the reviewer that NPL4 aggregation is likely not the only cytotoxic mechanism mediated by DSF-Cu++ administration. DNA damage and stem-cell targeting seem in fact to be involved in the anti-cancer effects of the combination, as demonstrated by several studies in CNS and non-CNS malignancies (1, 3, 4). In our study these mechanisms were confirmed by different tests, and we believe that they still partially contribute to the final cytotoxic and anticancer effect of this combination.

Were any major targets affected by NPL4 aggregation? Currently, there is no information on the role of NPL4 in therapy-targeted ubiquitin destruction of proteins. 

We thank the reviewer for this comment. As shown by Skrott et al. the major downstream effector of NPL4 aggregation resides in the activation of Heat-Shock Response proteins, with consequent triggering of cell apoptosis (1). As a mediator of the p97 cascade, NPL4 plays an indirect role in ubiquitin-mediated destruction of proteins.

How were the cell extracts prepared for NPL4 Western blots? Soluble or insoluble fractions used for SDS-PAGE? 

We thank the reviewer for pointing this out. Insoluble fractions were used for Western Blotting, and a more detailed description of the methods used for sample preparation is now included in the Supplementary files.

If aggregated, much of the protein is likely to remain with cell debris after lysis. Did authors find NPL4 aggregation in xenografted tumor specimens after Cu-DSF administration?

We thank the reviewer for this comment. 

WB showed a significant increase in NPL-4 signal after in vivo treatment with DSF-Cu++. A larger cohort, better timing of sample collection and the possible transient expression of NPL4 aggregation may explain this finding. Furthermore, while other groups demonstrated in vitro clustering of this mediator, in vivo evidence has been so far very limited and not definitive, somehow reinforcing the idea of the ephemeral action of NPL-4(1).

3) How do the DSF and Cu doses used in animal experiments compare with those used in clinical trials?

We thank reviewer for this comment. The DSF and Cu++ dosages used in mouse are significantly lower than the maximum tolerated dose in humans. Recent trials used doses of 125 mg/2 mg, twice daily, compared to our dosage of 150 mg/kg/day and 2 mg/kg/day (2). A case report in a Glioblastoma patient also reported a daily dosage of DSF and Cu++ of 250 and 6 mg, respectively (5). However, given the faster metabolic rate typical of rodents, preliminary studies in Glioblastoma and AT/RT used higher dosages than those administered to human patients. Specifically, 100mg/kg/day was used in Glioblastoma (4) and AT/RT (3). For the dosage used in our study, we did not determine any behavioral changes or body weight loss during our treatment.

FUNDING

The Funding Statement and Funding section in the Manuscript do not coincide because I was unable to include the Ms. Kimberly Spiro, the Donald W. Spiro Foundation, and Donald R. Spiro donations as a source of funding. No commercial interests are connected with this donation. I would like to include Ms. Kimberly Spiro, the Donald W. Spiro Foundation, and Donald R. Spiro donations in the final manuscript. Thank you

COMMERCIAL INTERESTS

The Commercial Interests statement in the manuscript is up to date.

1. Skrott Z, Mistrik M, Andersen KK, Friis S, Majera D, Gursky J, et al. Alcohol-abuse drug disulfiram targets cancer via p97 segregase adaptor NPL4. Nature. 2017;552(7684):194.

2. Huang J, DeWees T, Campian JL, Chheda MG, Ansstas G, Tsien C, et al. A TITE-CRM phase I/II study of disulfiram and copper with concurrent radiation therapy and temozolomide for newly diagnosed glioblastoma. American Society of Clinical Oncology; 2019.

3. Choi SA, Choi JW, Wang K-C, Phi JH, Lee JY, Park KD, et al. Disulfiram modulates stemness and metabolism of brain tumor initiating cells in atypical teratoid/rhabdoid tumors. Neuro-oncology. 2015;17(6):810-21.

4. Lun X, Wells JC, Grinshtein N, King JC, Hao X, Dang N-H, et al. Disulfiram when combined with copper enhances the therapeutic effects of temozolomide for the treatment of glioblastoma. Clinical Cancer Research. 2016;22(15):3860-75.

5. Karamanakos PN, Trafalis DT, Papachristou DJ, Panteli ES, Papavasilopoulou M, Karatzas A, et al. Evidence for the efficacy of disulfiram and copper combina-tion in glioblastoma multiforme-A propos of a case. Journal of BU ON: official journal of the Balkan Union of Oncology. 2017;22(5):1227-32.

---

## [Decision Letter · Decision Letter 1]

7 May 2021

Disulfiram and Copper Combination Therapy Targets NPL4, Cancer Stem Cells and Extends Survival in a Medulloblastoma Model

PONE-D-20-38226R1

Dear Dr. Serra,

We’re pleased to inform you that your manuscript has been judged scientifically suitable for publication and will be formally accepted for publication once it meets all outstanding technical requirements.

Kind regards,

Ilya Ulasov, Ph.D

Academic Editor

PLOS ONE

Additional Editor Comments (optional):

Reviewers' comments:

Reviewer's Responses to Questions

**Comments to the Author**

1. If the authors have adequately addressed your comments raised in a previous round of review and you feel that this manuscript is now acceptable for publication, you may indicate that here to bypass the “Comments to the Author” section, enter your conflict of interest statement in the “Confidential to Editor” section, and submit your "Accept" recommendation.

Reviewer #1: All comments have been addressed

2. Is the manuscript technically sound, and do the data support the conclusions?

Reviewer #1: Yes

3. Has the statistical analysis been performed appropriately and rigorously? 

Reviewer #1: Yes

4. Have the authors made all data underlying the findings in their manuscript fully available?

Reviewer #1: Yes

5. Is the manuscript presented in an intelligible fashion and written in standard English?

Reviewer #1: Yes

6. Review Comments to the Author

Reviewer #1: This is an emerging therapeutic area, particularly for brain cancers. It will further evoke interest in the concerned scientific community.

7. PLOS authors have the option to publish the peer review history of their article (what does this mean?). If published, this will include your full peer review and any attached files.

Reviewer #1: No

---

## [Editor Report · Acceptance letter]

26 Aug 2021

PONE-D-20-38226R1 

Disulfiram and Copper Combination Therapy Targets NPL4, Cancer Stem Cells and Extends Survival in a Medulloblastoma Model 

Dear Dr. Serra:

I'm pleased to inform you that your manuscript has been deemed suitable for publication in PLOS ONE. Congratulations! Your manuscript is now with our production department. 

Kind regards, 

on behalf of

Dr. Ilya Ulasov 

Academic Editor

PLOS ONE